# How curriculum delivery translates into entrepreneurial skills: The mediating role of knowledge of information and communication technology

Javed Iqbal[1], Xie Yi[1]*, Muhammad Azeem Ashraf[2]*, Ruihua Chen[1]*, Jin Ning[2], Shahnaz Perveen[3], Zahid Imran[4]

1 School of Education, Guangzhou University, Guangzhou, China, 2 Research Institute of Educational Science, Hunan University, Changsha, China, 3 Department of Education, Government Sadiq College Women University, Bahawalpur, Pakistan, 4 Lahore Business School, University of Lahore, Lahore, Pakistan

☯ These authors contributed equally to this work.
* r.chen@gzhu.edu.cn (RC); azeem20037@gmail.com (MAA); soexieyi@gzhu.edu.cn (XY)

**Data Availability Statement:** The data underlying the results presented in the study is attached in supporting information.

## Abstract

This research examines how curriculum delivery predicts entrepreneurial skills, with knowledge of information and communication technology (ICT) as a mediator. Curriculum delivery with the multiple dimensions of objectives, contents, teaching strategies, and feedback and assessment was used in this study, and a quantitative research design was adopted. A questionnaire survey was used to collect data from 482 students at six universities in Lahore, Pakistan, and the partial-least-squares structural equation model in SmartPLS 3.2 was used for data analysis. The results show that all dimensions of curriculum delivery (i) do not influence entrepreneurial skills and (ii) positively influence the knowledge of ICT. Also, in the indirect relationships, all dimensions of curriculum delivery (i.e., objectives, contents, teaching strategies, and feedback and assessment) are associated positively with ICT knowledge. Therefore, ICT knowledge plays a mediating role between curriculum delivery and entrepreneurial skills. The results also show that curriculum delivery for educational entrepreneurs is not working effectively and efficiently in Pakistani universities, and it is concluded that curriculum delivery and ICT knowledge boost entrepreneurial skills. Finally, the conclusions, limitations, and practical implications of this study are presented in detail.

## Introduction

The primary aim of higher education (HE) is to develop lifelong learners for the social good. Universities are the basis for the development and growth of countries in the contemporary global economic and technological world, and they propel economic growth by delivering curricula comprising advanced knowledge of technology and entrepreneurial skills (ES) [1, 2]. Therefore, this study reports the interventional effects of knowledge of information and communication technology (ICT) between curriculum delivery (CD) and ES to establish implications for instructors and administrators.

**Funding:** This study is supported by the National Natural Science Foundation of China (Grant No. 71950410624), The National Social Science Fund of China (Grant No. BHA210139), and Guangdong Planning Office of Philosophy and Social Sciences (Grant No. GD21YJY14). The funders had no role in study design, data collection and analysis, decision to publish, or preparation of the manuscript.

**Competing interests:** The authors have declared that no competing interests exist

CD is the antecedent of ES and is divided into four dimensions: (i) objectives, (ii) content and material, (iii) teaching strategies, and (iv) feedback and assessment. Given the importance of entrepreneurship to the economy [3], universities are devoted to designing curricula that develop ES in their students [3–6]. Integrated CD across disciplines is now considered as the engine to create employment opportunities, especially among young people in various countries [7]. Pakistan is also well aware of the challenges of the youth population and unemployment, and to account for those challenges, its universities must revise and integrate their entrepreneurial CD in cross-disciplinary teaching in Pakistan [8].

Many studies have discussed the significant role of entrepreneurial education in producing ES for economic benefits [9–11]. ES require students to put theory into practice and become proactive and creative, and they help students to become skilled at critical thinking, problem solving, risk taking, and innovation. Moreover, entrepreneurial teaching supports students in learning new ideas and new technology practices in classes [12]. However, we could find very few studies that explored the relationship between ES and ICT knowledge, hence the present study that does so.

Different sectors—including education—have changed their processes via ICT, and ICT knowledge has transformed the approaches of teaching, learning, and communication [13]. The integration of ICT knowledge in entrepreneurial education is seen as evidence of successful entrepreneurs in the western world [14], and ICT knowledge and innovation facilitate the development of ES in universities [13]. Teachers must have ICT knowledge and deliver the curriculum effectively (e.g., objectives, content and material, teaching strategies, and feedback and assessment), and ICT knowledge might be used to develop ES [15]. Based on the above discussion, it is assumed that ICT knowledge and CD are the antecedents of ES. Therefore, this study explores the mediating role of ICT knowledge between CD and ES. To the best of our knowledge, this is the first study conducted in Pakistan on the role of ICT knowledge between CD and ES. This study contributes to the interface among CD, ICT knowledge, and entrepreneurship in university education. First, this study adds important practical knowledge to the insufficient understanding of how CD and ICT knowledge actually affect the process of ES (critical thinking, risk taking, problem solving, and innovation) development in universities. Second, based on the findings, multiple suggestions and implications are established for administrators and policymakers regarding how they could plan for developing the ES of students. Therefore, the aim of the study was to answer the following questions.

**Research question 1:** How do CD (objectives, content and material, teaching strategies, and feedback and assessment) and ICT knowledge influence the ES of students (prospective educators)?

**Research question 2:** How does ICT knowledge mediate the relationship between CD (objectives, content and material, teaching strategies, assessment and feedback) and the ES of students (prospective educators)?

## Literature review

### Background of entrepreneurial curriculum delivery in Pakistan

Universities are becoming well aware of entrepreneurial CD and its importance worldwide, and CD is vital for developing ES among university students [3]. In Pakistan, many scholars have discussed the practical and implementation issues of CD. Tanveer and Haq [8] described the issues associated with an integrated curriculum and its delivery in Pakistani universities. The Higher Education Commission (HEC) of Pakistan is now taking some measures to redesign the integrated entrepreneurial curriculum across disciplines [8, 16]. The HEC introduced

faculty exchange programs with top-ranking international universities to develop competent and qualified faculty staff for effective entrepreneurial CD through teaching. Since then, the CD has been applied extensively in Pakistani universities to develop educational entrepreneurs [8]. However, the traditional system of teaching may not necessarily support implementing an integrated curriculum across disciplines. Centered on the issues raised above, this study emphasizes measuring how CD affects the ES of students.

## Curriculum delivery

CD involves objectives, contents, teaching strategy, and feedback and assessment to fulfill the needs of all students. Curriculum objectives are the expected pedagogical, social, economic, or combined outcomes, whereas the contents are the thematic areas covered by the courses. Teaching strategy refers to delivering the course contents to facilitate understanding of the core concepts and learning as per the course objectives. Feedback and assessment refer to measuring the extent to which the CD has been successful [17]. Feedback and assessment are generally driven by the need to evaluate progress (formative assessment) and judge performance (summative assessment). CD is adapted according to the nature of the course, especially regarding ES. Acquiring ES successfully depends on multiple factors, but the theory of planned behavior (TPB) considers the most critical one to be quality education via successful CD [18]. Students can learn ES by condensing the benefits of effective CD in universities. Previous studies have also discussed the dimensions of CD (objectives, teaching strategies, contents and material, and feedback assessment), arguing that these contribute to the ES of students [3, 19]. Therefore, CD is becoming essential for developing ES among students in universities, hence the present study is focused on CD.

## Entrepreneurial skills

Organizations pay the price when their human resources lack ES [20], thereby stifling their income resources [21]. ES is further defined as practical know-how to successfully establish or run a business [22]. Dahlstrom and Talmage [20] defined ES as the talent to understand the market opportunities and construct a solution to capture those opportunities to start activities. In line with this, the present study defines ES as the abilities of critical thinking, problem solving, risk taking, and innovation [12]. Critical thinking is the process of encouraging students to move from "knowing" to "thinking," thereby cultivating advanced thinking skills that are compulsory for success in entrepreneurial careers. Problem solving is the practice of discovering explanations for difficult or complex issues during work activities. Risk taking is the ability to be ready for undesirable results of any action taken to start a profession. Innovation is the process whereby an entrepreneur puts into practice an idea that results in a new service [23]. The contents related to these ES must incorporate designing curricula through further university entrepreneurial education to produce successful entrepreneurs [24]. Modern university education provides students with an extensive set of ES to produce a positive entrepreneurial approach concerning the start of a new professional venture [25]. With effective CD, ICT knowledge is also considered a predictor of developing ES [17].

## ICT knowledge

ICT emphasizes the roles of unified communication [26] and the integration of telecommunications devices (e.g., smartphones) and applications as necessary enterprise software that supports workers to access, store, spread, and operate information [27]. ICT knowledge products and applications can enhance the cognitive abilities necessary for ES [28]. ICT knowledge is how to use ICT tools in developing ES among educational entrepreneurs. It increases

productivity and effectiveness and reduces costs [15]. Much research has been conducted to explore the effectiveness of ICT knowledge in entrepreneurial education [29, 30]. In this study, we define ICT knowledge as (i) the management of information technology, customized software and applications, and websites, (ii) the knowledge of technological information, and (iii) the communication of information data, and we measure the mediating role of ICT knowledge between EC and ES.

## Hypotheses development and theoretical framework

In this study, the proposed model highlights that CD can increase ES in students through the ICT knowledge of teachers. CD and ICT knowledge have long been major concerns among researchers and practitioners for their role in developing ES. CD in universities is considered the antecedent of ES [31], and the TPB contends that CD may make an important difference to ES [32]. In dynamic university environments, CD impacts students' ideas and encourages them to be entrepreneurs. Therefore, we contend that ICT knowledge mediates the connection between CD and ES. The current study analyzes these relationships empirically and highlights the impact of CD on ES through ICT knowledge in a developing nation. It is well established that students can learn ES by being involved in the curricular activities in university entrepreneurial education. ES has been defined as critical thinking, problem solving, risk taking, and innovation [12]. However, ES might also be evaluated concerning the university ecosystem. It is now common practice for universities to painstakingly analyze their CD effectiveness against their market needs to help students learn ES [3, 31]. It is challenging for universities to produce quality entrepreneurs through effective CD, but an inclusive strategy can enable universities to face this challenge [17, 33]. ES should be aligned with the curriculum objectives to make ES important for student success, otherwise students will be unable to start their work activities, thereby making them unsuccessful entrepreneurs. The existing literature explains that CD has a significant relationship with ICT knowledge and ES for students [14, 15], and the key purpose of CD is to develop ES through ICT knowledge among university students [13].

However, universities are also in dire need of designs that integrate curricula across disciplines, and those designs must involve critical thinking, problem solving, risk taking, and innovation and develop confidence in integrating curricula for successful entrepreneurs [12]. Based on the TPB, some scholars have claimed that students' perception of effective university education (CD) has deep-rooted influences on their ES [19, 32, 34]. Furthermore, students' critical thinking, problem solving, risk taking, and innovation abilities are to be considered the indicators of ES, thereby helping them to become successful entrepreneurs. Based on these ideas, we present the proposed research model in Fig 1, which exhibits all of the hypotheses.

## Curriculum delivery and entrepreneurial skills

Numerous curriculum and entrepreneurship studies have shed light on the positive association between CD (objectives, contents and material, teaching strategies, and feedback and assessment) and ES [3, 17, 35], and there have been many studies of effective CD and how it affects ES, also confirming a positive association [36–38]. The TPB also supports the notion that entrepreneurial education (curriculum) is very important for ES, and curriculum is key among all factors of entrepreneurial education [39, 40]. Furthermore, the TPB states that ICT positively affects ES [18, 19, 40]. Some scholars with a common approach have argued that CD collectively develops ES, and they have recommended that universities must redesign ES [8]. Others have suggested that specific forms of effective CD dimensions as objectives, contents, materials, teaching strategies, and feedback and assessment result in students' ES [17]. These findings have helped greatly in understanding the relationships between CD and its

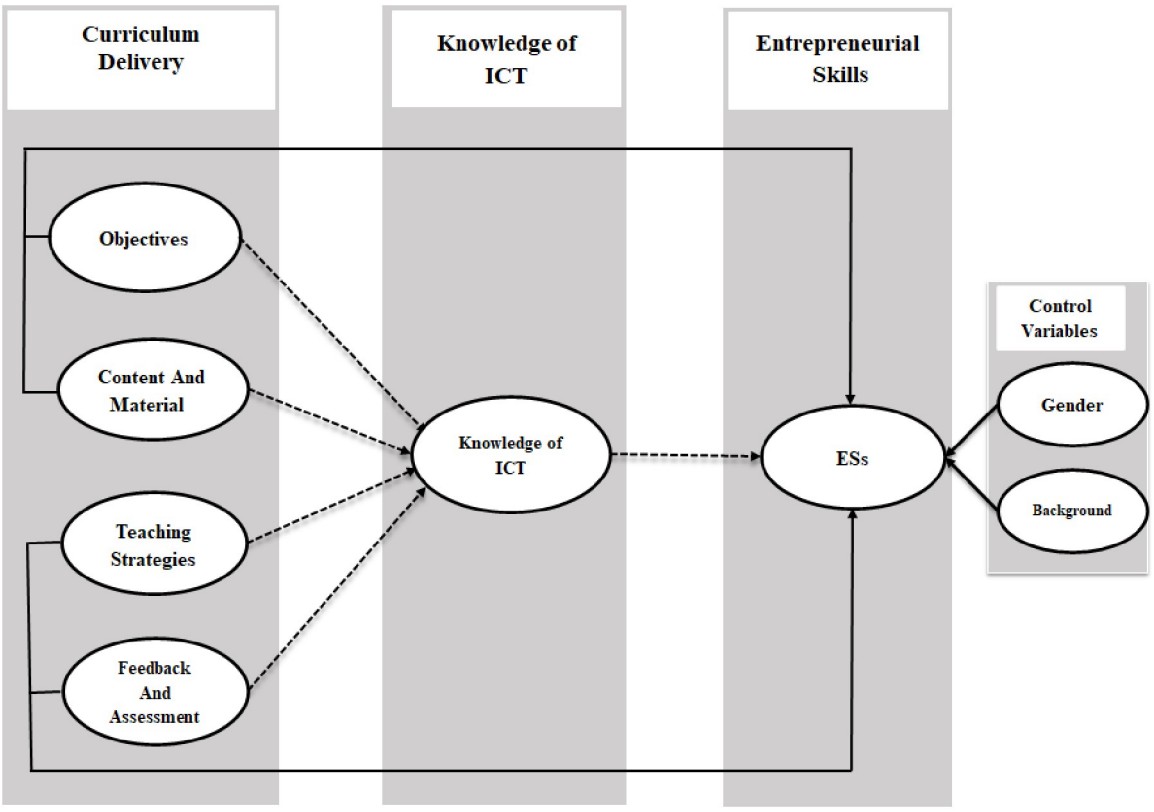

**Fig 1. Research model.** Solid and dashed arrows correspond to direct and indirect links, respectively, in the model.

dimensions and ES. Thus, a positive relationship between CD and its sub-dimensions and ES is assumed with the following hypotheses.

**H1a:** Curriculum objectives have a positive effect on ES.

**H1b:** Curriculum contents and materials have a positive effect on ES.

**H1c:** Curriculum teaching strategies have a positive effect on ES.

**H1d:** Curriculum feedback and assessment have a positive effect on ES.

## Curriculum delivery and ICT knowledge

Many scholars have developed theoretical models and concepts to bring out the relationship between CD (objectives, contents and materials, teaching strategies, feedback and assessment) and ICT knowledge. There is clear evidence that CD knowledge has a positive relationship with ICT knowledge [41–43], and the literature argues consistently that effective CD in universities enhances ICT knowledge [44, 45]. Also, integrated CD across disciplines increases interest among students to learn ICT knowledge. CD may become effective with updated contents and materials and modern teaching strategies that help students to learn ICT knowledge [45, 46]. These arguments suggest that each of the three forms of CD is positively related to ICT knowledge, which leads to the following hypotheses.

**H2a:** Curriculum objectives have a positive effect on ICT knowledge.

**H2b:** Curriculum contents have a positive effect on ICT knowledge.

**H2c:** Curriculum teaching strategies have a positive effect on ICT knowledge.

**H2d:** Curriculum feedback and assessment have a positive effect on ICT knowledge.

## ICT knowledge and entrepreneurial skills

Multiple studies have presented theoretical models and concepts for establishing the relationship between ICT knowledge and ES, and there is clear evidence that ICT knowledge has a positive relationship with ES [14, 47, 48]. The literature argues consistently that innovation in entrepreneurial education affects critical-thinking, risk-taking, problem-solving, and innovation skills [12], as does the adoption of new ICT knowledge such as (i) the management of information technology, customized software and applications, and websites, (ii) the knowledge of technological information, and (iii) the communication of information data [27, 28]. ICT knowledge is a strong antecedent of ES, which suggests that ICT knowledge is positively related to ES and leads to the following hypothesis.

**H3:** ICT knowledge has a positive effect on ES.

## Mediating effect of ICT knowledge

Most of the studies reviewed above indicate that ICT knowledge has a direct effect on ES. However, Hynes and Richardson [49] suggested that ICT knowledge plays a positive role in ES and confirmed that CD enhances ES. Barnett et al. [28] measured the role of ICT with information and knowledge acquisition to increase ES. Moreover, Duruamaku-Dim and Duruamaku-Dim [50] concluded that ICT is highly relevant in entrepreneurship education and provides support for enriching the teaching and learning process to enhance the ES and entrepreneurial attitudes among graduates, and they suggested that this phenomenon is yet to be explored. Portuguez Castro and Gómez Zermeño [13] highlighted the significance of an e-learning educational model that can contribute to the development of an entrepreneurial spirit and competencies among graduates. However, we can find no study that describes the mediating role of ICT knowledge between CD and ES, so we propose the following hypotheses.

**H4a:** ICT knowledge plays a mediating role between curriculum objectives and ES.

**H4b:** ICT knowledge plays a mediating role between curriculum contents and ES.

**H4c:** ICT knowledge plays a mediating role between curriculum teaching strategies and ES.

**H4d:** ICT knowledge plays a mediating role between curriculum feedback and assessment and ES.

## Research methods

The HE sector of Pakistan as a developing country was selected for the present study for the following reasons. Most research on this area has been conducted in advanced countries, with comparatively less research conducted in developing countries with different curriculum aspects. To date, Pakistan has established private and public education systems through educational entrepreneurs, which motivates this research. Effective CD, ICT knowledge, and ES among prospective educators could play a vital role in developing new educational organizations in Pakistan. The HE sector is a useful tool for developing ES among students and is a good context for this research, being readily available to us.

## Research approach

A questionnaire survey was used for this research, this being a common method for a wide range of empirical responses. This design is pervasive for generating valid and reliable findings

in research of this type [51, 52]. We adopted the questionnaire as an instrument for primary quantitative data collection from representatives of the target population. This study is part of a larger project focused on the impact of the Internet and technologies in HE settings, and the ethics committee of Hunan University approved the study. Its aims and objectives were sent to the participating universities prior to their approval for data collection, and all ethical considerations were followed throughout the research process.

## Instrument development

Thirty-eight items were included in the questionnaire, each offering respondents choices from a seven-point Likert scale. The 22 items related to CD (objectives, contents, teaching methodology, and feedback and assessment) were adopted and modified from the work of TLE-MU-HKU [53], the seven items related to ICT knowledge were adopted and modified from the work of Gulbahar and Guven [54] and Tippins and Sohi [55], and the nine items related to ES were adopted and modified from the work of Man [56] and Li [57]. To assess the reliability and validity of the instrument before data collection, a pilot study was conducted with 60 participants who had similar demographic characteristics to those of the final sample. The respondents were asked to highlight any errors and to suggest any modifications required in the instrument items. The corrections and modifications were made accordingly, ensuring that all items were well written and that the participants responded to the questionnaire successfully. The details of the measurement (research questionnaire) and factor loading of each item are presented in Table 2.

## Measures

**Curriculum objectives.**   The statements related to CD (objectives) were adopted and modified from the work of TLEMU-HKU [53]. Three statements were used on a seven-point Likert scale (1 = strongly disagree, 7 = strongly agree), with sample statements such as "The teachers make it clear right from the start that they expect us to become educational entrepreneurs" and "The teacher provides detailed information about information technology capabilities at the beginning of the semester." The Cronbach's alpha for CD (objectives) and the standard Cronbach's alpha index (0.749) confirm that the construct was considered reliable (see Table 2).

**Contents and materials.**   The statements related to CD (contents and materials) were adopted and modified from the work of TLEMU-HKU [53]. Seven statements were used on a seven-point Likert scale (1 = strongly disagree, 7 = strongly agree), with sample statements such as "I have acquired sufficient content knowledge about entrepreneurship" and "I have various alternative ways for developing my understanding of information technology." The Cronbach's alpha for CD (contents and materials) was 0.802 (see Table 2), and the standard Cronbach's alpha index is 0.70, so the CD (contents and materials) construct was considered reliable.

**Teaching strategies.**   The statements related to CD (teaching strategies) were adopted and modified from the work of TLEMU-HKU [53]. Six statements were used on a seven-point Likert scale (1 = strongly disagree, 7 = strongly agree), with sample statements such as "Our teachers keep me motivated and engaged in entrepreneurial learning" and "Our teachers have a meaningful interaction between students and teachers beyond the classroom." The Cronbach's alpha for CD (teaching strategies) was 0.894, and the standard Cronbach's alpha index is 0.70, so the CD (teaching strategies) construct was considered reliable (see Table 2).

**Feedback and assessment.**   The statements related to CD (feedback and assessment) were adopted and modified from the work of TLEMU-HKU [53]. Six statements were used on a

seven-point Likert scale (1 = strongly disagree, 7 = strongly agree), with sample statements such as "Our teachers give constructive feedback on my progress" and "The teachers invest much time into commenting on my work." The Cronbach's alpha for CD (feedback and assessment) was 0.890 (see Table 2), and the standard Cronbach's alpha index is 0.70, so in this research, the CD (feedback and assessment) construct was considered reliable.

**ICT knowledge.** The statements related to ICT knowledge were adopted and modified from the work of Tippins and Sohi [55]. Seven statements were used on a seven-point Likert scale (1 = strongly disagree, 7 = strongly agree), with sample statements such as "I know about information technologies that I can use for understanding the entrepreneurial process" and "I know how to use information systems that help redesign the institutional processes in the school." The Cronbach's alpha for ICT knowledge was 0.873 (see Table 2), and the standard Cronbach's alpha index is 0.70, so in this research, the ICT knowledge construct was considered reliable.

**Entrepreneurial skills.** The statements related to ES were adopted and modified from the work of Tippins and Sohi [55]. Nine statements were used on a seven-point Likert scale (1 = strongly disagree, 7 = strongly agree), with sample statements such as "After completing my education program, I can make a rational decision in the school" and "After completion of my education program, I would negotiate with others." The Cronbach's alpha for ES was 0.892, and the standard Cronbach's alpha index is 0.70, so the ES construct was considered reliable (see Table 2).

## Data collection

The data were collected from 10 universities—five private and five public sector—that provide education to educational entrepreneurs in the city of Lahore in the province of Punjab, Pakistan. Prior to the data collection, the aims and objectives of the study were sent to the leaders of the universities to ask for permission to collect data. All participants were provided with a consent form prior to filling in the questionnaire, and they were informed about the aims and objectives of the research, and that their identity will not be disclosed at any stage of the research. The respondents were undergraduate, graduate (Masters or equivalent), and post-graduate (Doctoral or equivalent) students (prospective educational entrepreneurs who were enrolled in educational leadership and management programs) who aspired to become educational entrepreneurs in the future. We circulated 600 questionnaires among students of the 10 universities, of which 520 were returned, of which 38 were rejected because of incomplete data. The final sample comprised 482 responses with a return rate of 80.33%. In response to privacy concerns from the respondents, we marked the selected universities alphabetically from A to J (see Fig 2).

## Demographics

Produced using SPSS 22, Table 1 gives the demographic characteristics of the sample, which comprised 125 male (25.9%) and 357 female (74.1%) respondents, of whom 204 were from rural areas (42.3%) and 278 were from urban areas (57.7%). Table 1 shows that the respondents were studying at various levels, i.e., 163 undergraduates (33.8%), 278 graduates (55.8%), and 50 postgraduates (10.4%). The data show the distribution of cumulative grade point average (CGPA) of the respondents, with three having a CGPA of less than 2.00 (0.6%) 128 having a CGPA of 2.00–3.00 (26.6%), 244 having a CGPA of 3.10–3.50 (50.6%), and 104 having a CGPA of 3.51–4.00 (21.6%). The study sample also comprised 334 (69.3%) and 148 (30.7%) respondents from public and private universities, respectively. Table 1 summarizes the demographics of the respondents.

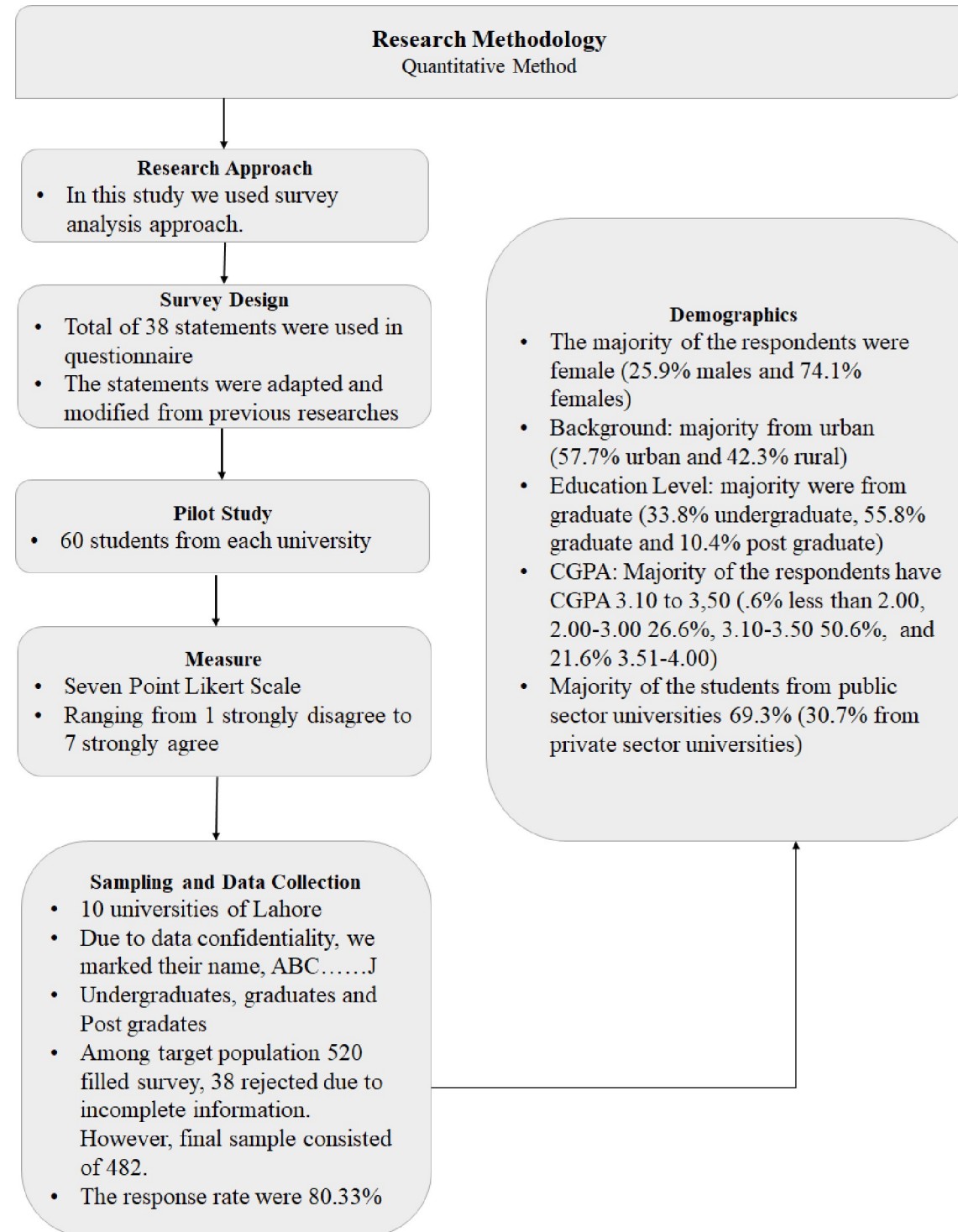

**Fig 2. Methodology chart.**

**Table 1. Demographic characteristics of respondents.**

| Measure | Items | Frequency [*n*] | Percentage [%] |
|---|---|---|---|
| Gender | Male | 125 | 25.9 |
| | Female | 357 | 74.1 |
| | Total | 482 | 100.0 |
| Background | Rural | 204 | 42.3 |
| | Urban | 278 | 57.7 |
| | Total | 482 | 100.0 |
| Education level | Undergraduate | 163 | 33.8 |
| | Graduate | 269 | 55.8 |
| | Post-graduate | 50 | 10.4 |
| | Total | 482 | 100.0 |
| CGPA | <2.00 | 3 | 0.6 |
| | 2.00–3.00 | 128 | 26.6 |
| | 3.10–3.50 | 244 | 50.6 |
| | 3.51–4 | 104 | 21.6 |
| | Total | 482 | 100.0 |
| Sectors | Public | 334 | 69.3 |
| | Private | 148 | 30.7 |
| | Total | 482 | 100.0 |

# Analysis and results

## Preliminary considerations

Partial least squares (PLS) structural equation modeling (SEM) calculates partial model structures by merging principal-component analysis with ordinary least-square regressions. We used composite path models to test the theoretical framework. PLS-SEM is a dissimilar technique from other techniques for examining a prediction perspective and deals with structural models that include several constructs, indicators, and model relationships. A path model includes measured and reflective constructs that may have inclusive substantiation in measurement theory, and it deals with small sample size. However, PLS-SEM works fine with large sample sizes. We chose PLS-SEM to deal with distributional assumptions, such as lack of normality, because the study involves latent variable scores for supplementary analyses. This offers an outline of when to consider determining whether PLS is a suitable SEM technique for social science research [58]. The high statistical command characteristics of PLS-SEM are very helpful in exploratory and confirmatory research aimed at examining less-developed theories. The main focus of PLS-SEM is the relationship between prediction and theory testing, wherein results must be validated [59]. Given the background, researchers have recently suggested new evaluation procedures that are proposed explicitly for the prediction-oriented nature of PLS-SEM [60].

## Measurement model

The evaluation of results in PLS-SEM mainly involves studying the measurement models, wherein appropriate criteria vary for reflective and formative constructs (see S1 File). It is considered necessary to evaluate the structural model if the required criteria are met by the measurement models [61]. Similar to other statistical techniques, PLS-SEM methods follow rules of thumb that function as guidelines to assess model results [62]. These rules of thumb suggest how the results should be interpreted, and they usually differ depending on the context. For

**Table 2. Construct reliability and convergent validity.**

| Constructs | IL | CA | rho_A | CR | AVE |
|---|---|---|---|---|---|
| Curriculum objectives (CO) | | 0.749 | 0.761 | 0.856 | 0.666 |
| CO1 | 0.847 | | | | |
| CO2 | 0.842 | | | | |
| CO3 | 0.756 | | | | |
| Contents and materials (CM) | | 0.802 | 0.806 | 0.855 | 0518 |
| CM1 | 0.627 | | | | |
| CM2 | 0.705 | | | | |
| CM3 | 0.689 | | | | |
| CM4 | 0.684 | | | | |
| CM5 | 0.605 | | | | |
| CM6 | 0.716 | | | | |
| CM7 | 0.700 | | | | |
| Teaching strategies (TS) | | 0.894 | 0.895 | 0.919 | 0.653 |
| TS1 | 0.791 | | | | |
| TS2 | 0.819 | | | | |
| TS3 | 0.828 | | | | |
| TS4 | 0.828 | | | | |
| TS5 | 0.797 | | | | |
| TS6 | 0.787 | | | | |
| Feedback and assessment (FA) | | 0.890 | 0.894 | 0.916 | 0.645 |
| FA1 | 0.808 | | | | |
| FA2 | 0.844 | | | | |
| FA3 | 0.800 | | | | |
| FA4 | 0.761 | | | | |
| FA5 | 0.773 | | | | |
| FA6 | 0.828 | | | | |
| Knowledge of information communication technology (KICT) | | 0.873 | 0.887 | 0.902 | 0.569 |
| KICT1 | 0.716 | | | | |
| KICT2 | 0.631 | | | | |
| KICT3 | 0.763 | | | | |
| KICT4 | 0.840 | | | | |
| KICT5 | 0.698 | | | | |
| KICT6 | 0.788 | | | | |
| KICT7 | 0.813 | | | | |
| Entrepreneurial skills (ES) | | 0.892 | 0897 | 0.912 | 0.535 |
| ES1 | 0.703 | | | | |
| ES2 | 0.690 | | | | |
| ES3 | 0.745 | | | | |
| ES4 | 0.755 | | | | |
| ES5 | 0.737 | | | | |
| ES6 | 0.712 | | | | |
| ES7 | 0.763 | | | | |
| ES8 | 0.797 | | | | |
| ES9 | 0.769 | | | | |

**Abbreviations:** IL, indicator loading; CA, Cronbach's alpha; CR, composite reliability; AVE, average variance extracted

**Table 3. Discriminant validity (HTMT).**

| Scales | CM | CO | ES | FA | KICT | TS |
|---|---|---|---|---|---|---|
| Content and material (CM) | 0.676 | | | | | |
| Curriculum objectives (CO) | 0.360 | 0.816 | | | | |
| Entrepreneurial skills (ES) | 0.398 | 0.834 | 0.732 | | | |
| Feedback and assessment (FA) | 0.440 | 0.439 | 0.476 | 0.803 | | |
| Knowledge of ICT (KICT) | 0.491 | 0.496 | 0.58 | 0.609 | 0.754 | |
| Teaching strategies (TS) | 0.545 | 0.466 | 0.469 | 0.655 | 0.568 | 0.808 |

example, exploratory research requires a reliability of 0.60 or higher, while research with established measures must have a reliability of 0.70 or higher. The last step involves applying one or more robustness checks to support the stability of the results when interpreting PLS-SEM results. Also, confirmatory factor analysis is used to measure the convergent and discriminant validity of the four constructs and estimate the fitness of the overall measurement model. The model fitness level must improve up to the proposed levels. We removed the items (having low indicator loading) from reflective constructs to achieve an acceptable scale index [58], and the standard reliability index was higher than 0.70.

We also measured all constructs according to the standard reliability value. Table 2 gives the Cronbach's alpha, rho_A, and composite reliability values of the variables, which are all higher than 0.70. The average variance extracted (AVE) values are all higher than 0.50. The measurement-model assessment includes investigating the indicator loadings: all constructs show an indicator loading of more than 0.60, which means that questionnaire has acceptable item reliability (see Table 2).

## Discriminant validity

Furthermore, Table 3 gives the heterotrait–monotrait (HTMT) value for discriminant validity. All constructs have HTMT < 0.85 (Table 3), which is significantly lower than the threshold value.

## Descriptive statistics

Table 4 presents the descriptive statistics of the respondents, whose responses were measured on a seven-point Likert scale. The mean values for all responses were in the range of 4.61–5.49, and the standard deviations were in the range of 0.947–1.516.

## Regression analysis

PLS-SEM works through a variance-based SEM technique that allows concurrent evaluation of the measurement model (e.g., evaluating the reliability and validity of measures in

**Table 4. Descriptive statistics.**

| Variables | N | Min | Max | Mean | SD |
|---|---|---|---|---|---|
| Curriculum objectives | 482 | 1 | 7 | 4.61 | 1.516 |
| Curriculum and material | 482 | 1 | 7 | 4.78 | 1.119 |
| Teaching strategies | 482 | 1 | 7 | 4.99 | 1.388 |
| Feedback and assessment | 482 | 1 | 7 | 5.13 | 1.221 |
| ICT knowledge | 482 | 1 | 7 | 4.99 | 1.625 |
| Entrepreneurial skills | 482 | 1 | 7 | 5.49 | 0.947 |

**Abbreviations:** N, number; Min, minimum; Max, Maximum; SD, standard deviation

**Table 5. Direct and indirect relations.**

| Direct effects | Coefficient | Mean | SD | T statistic | P value | Result |
|---|---|---|---|---|---|---|
| CO -> ES | 0.709 | 0.708 | 0.029 | 24.354 | 0.000 | Sig |
| CM -> ES | 0.047 | 0.046 | 0.029 | 1.611 | 0.107 | Insig |
| TS -> ES | -0.034 | -0.034 | 0.035 | 0.960 | 0.337 | Insig |
| FA -> ES | 0.045 | 0.045 | 0.040 | 1.132 | 0.258 | Insig |
| CO -> Knowledge of ICT | 0.212 | 0.214 | 0.046 | 4.624 | 0.000 | Sig |
| CM -> Knowledge of ICT | 0.186 | 0.192 | 0.043 | 4.305 | 0.000 | Sig |
| TS -> Knowledge of ICT | 0.146 | 0.143 | 0.054 | 2.726 | 0.006 | Sig |
| FA -> Knowledge of ICT | 0.338 | 0.338 | 0.056 | 5.983 | 0.000 | Sig |
| Knowledge of ICT -> ES | 0.199 | 0.202 | 0.040 | 4.936 | 0.000 | Sig |
| **Direct effects** | | | | | | |
| TS -> Knowledge of ICT -> ES | 0.029 | 0.029 | 0.012 | 2.447 | 0.014 | Sig |
| CM -> Knowledge of ICT -> ES | 0.037 | 0.039 | 0.012 | 3.210 | 0.001 | Sig |
| CO -> Knowledge of ICT -> ES | 0.042 | 0.044 | 0.014 | 2.924 | 0.003 | Sig |
| FA -> Knowledge of ICT -> ES | 0.067 | 0.068 | 0.016 | 4.150 | 0.000 | Sig |
| Gender -> ES | 0.047 | 0.047 | 0.025 | 1.870 | 0.061 | Insig |
| Background -> ES | 0.041 | 0.041 | 0.026 | 1.587 | 0.113 | Insig |

**Abbreviations:** *SD, standard deviation; Sig, significant value, p < .05; Insig, Insignificant*

conceptual variables) and the structural model (e.g., examining the important relationships between constructs included in the model) [58, 63, 64]. The path coefficient significance was tested by applying PLS-SEM bootstrapping algorithm path analysis method. The number of sub-samples for the bootstrap was 5000, which is widely accepted. Table 5 shows the direct effects of the four constructs of CD (objectives, contents, teaching strategies, and feedback and assessment) on ES and the indirect effect of ICT knowledge between CD and ES. The empirical analysis results indicate that the relationship between variables is insignificant except between ICT knowledge and ES in the final model (Fig 3).

In **H1a**, the study assumes that curriculum objectives have a significant and positive effect on ES, and Table 5 and Fig 3 show that curriculum objectives influence ES significantly ($\beta = 0.709$, $T = 24.354$, $p < 0.05$); therefore, **H1a** is supported. In **H1b**, the study assumes that curriculum contents and materials have a significant and positive influence on ES, but Table 5 and Fig 3 show that curriculum contents and materials influence ES insignificantly ($\beta = 0.047$, $T = 1.611$, $p > 0.05$); therefore, **H1b** is not supported. In **H1c**, the study assumes that curriculum teaching strategies have not a significant and positive influence on ES, but Table 5 and Fig 3 show that curriculum teaching strategies influence ES insignificantly ($\beta = -0.034$, $T = 0.960$, $p > 0.05$); therefore, **H1c** is not supported. In **H1d**, the study assumes that curriculum feedback and assessment have a significant and positive effect on ES, and Table 5 and Fig 3 revealed that curriculum feedback and assessment influence ES significantly ($\beta = 0.45$, $T = 1.132$, $p > 0.05$); therefore, **H1d** is not supported.

In **H2a**, the study assumes that CD (objectives) has a significant and positive effect on ICT knowledge, and Table 5 and Fig 3 show that CD (objectives) influences ICT knowledge significantly ($\beta = 0.212$, $T = 4.624$, $p < 0.05$); therefore, **H2a** is supported. In **H2b**, the study assumes that CD (contents and materials) has a significant and positive effect on ICT knowledge, and Table 5 and Fig 3 show that CD (contents and materials) influences ICT knowledge significantly ($\beta = 0.186$, $T = 4.305$, $p < 0.05$); therefore, **H2b** is supported. In **H2c**, the study assumes that CD (teaching strategies) influences ICT knowledge significantly and positively, and

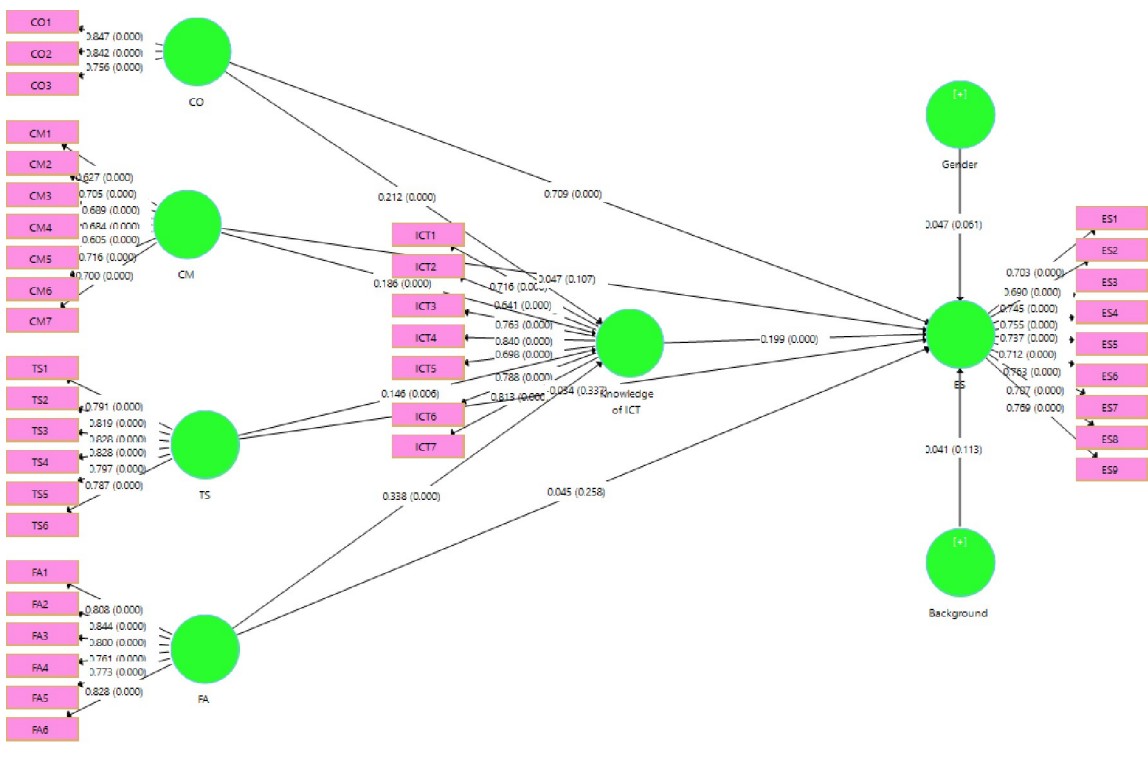

**Fig 3. PLS-SEM.**

Table 5 and Fig 3 show that CD (teaching strategies) influences ICT knowledge significantly ($\beta$ = 0.146, $T$ = 2.726, $p < 0.05$); therefore, **H2c** is supported. In **H2d**, the study assumes that CD (feedback and assessment) has a significant and positive influence on ICT knowledge, and Table 5 and Fig 3 show that CD (feedback and assessment) influences ICT knowledge significantly ($\beta$ = 0.338, $T$ = 5.983, $p < 0.05$); therefore, **H2d** is supported. In **H3**, the study assumes that ICT knowledge has a significant and positive influence on ES, and Table 5 and Fig 3 show that ICT knowledge influences ES significantly ($\beta$ = 0.199, $T$ = 4.936, $p < 0.05$); therefore, **H3a** is supported. Moreover, in indirect relationships, ICT knowledge mediates between ES and curriculum objectives ($\beta$ = 0.029, $T$ = 2.447, $p < 0.05$), curriculum contents and materials ($\beta$ = 0.037, $T$ = 3.210, $p < 0.05$), curriculum teaching strategies ($\beta$ = 0.042, $T$ = 2.924, $p < 0.05$), and curriculum feedback and assessment ($\beta$ = 0.067, $T$ = 4.150, $p < 0.05$). As noted in Table 5 and Fig 3, hypotheses **H4a–H4d** are supported. Moreover, we measured two controlled variables such as Gender and Background of the participants, of both of them no variable has direct and significant positive effect on ES ($\beta$ = 0.047, $T$ = 1.870, $p > 0.05$) ($\beta$ = 0.041, $T$ = 1.587, $p > 0.05$).

## Discussion

The majority of research on CD has been conducted in advanced countries. This study has added to the literature for curriculum policymakers, instructors, and administrators in Pakistan. First, it examined the influence of CD on ES and ICT knowledge. Second, it investigated the influence of ICT knowledge on ES in the Pakistani HE sector. Third, it explored the mediating role of ICT knowledge between CD and ES.

This study began by investigating the direct relationship between CD (objectives, contents, teaching strategies, and feedback and assessment) and ES. The results showed that CD

(curriculum objectives) influences ES significantly, which supports our insights in hypotheses **H1a**. Previous research showed that CD influences ES positively [3, 17]. Correspondingly, Boldureanu et al. [65] conducted a study on university students and found that an entrepreneurship curriculum plays a positive role regarding ES. Therefore, it is concluded that effective CD is the antecedent of ES among students. Similarly, curriculum content, teaching strategies, and feedback and assessment have insignificant associations with ES, which do not support our hypotheses **H1b–H1d**. The results of the present study are consistent with the outcomes of previous studies [36]. Additionally, Reyad et al. [23] found that an entrepreneurial curriculum was not enhancing student ES such as risk taking, critical thinking, problem solving, and innovation. Therefore, the results of the present study may be due to the weak curriculum content and material, teaching strategies, and evaluation approaches, and so CD requires some changes.

Second, this research explored the positive and direct effect of CD (contents, teaching strategies, and feedback and assessment) on ICT knowledge and supported our hypotheses **H2a–H2d**. Moreover, previous studies showed that CD has a positive and direct relationship with ICT knowledge [66, 67]. Fetter [68] studied the US health sector and found that curriculum strategies improve ICT learning outcomes. Therefore, it seems possible that CD is substantial for student ICT capabilities and knowledge.

Third, this study explored the positive and significant direct relationship between ICT knowledge and ES, supporting our hypothesis **H3**. Previous research has provided evidence that ICT knowledge is associated positively and directly with ES [69, 70]. Chatterjee et al. [71] studied female empowerment in India as a developing country and explored how ICT adoption enhances ES. Ultimately, ICT knowledge facilitates ES.

Fourth, this study explored the mediating role of ICT knowledge between CD (contents, teaching strategies, feedback and assessment) and ES, supporting our hypotheses **H4a–H4d**. In our literature review, we found some research on the mediation of ICT knowledge between these variables, but none examining the mediating role of ICT knowledge between CD and developing ES. Therefore, this aspect is original to the present research.

## Conclusions

The findings of the present research have substantial implications for curriculum designers and developers. This study accelerates the debate on collectively examining CD, ES, and ICT capabilities. It explored the direct relationship of CD on ES and ICT capabilities and investigated the mediating role of ICT knowledge between CD and ES, finding relationships among CD, ES, and ICT knowledge.

The results show that (i) in the direct relationships, CD (curriculum objectives) influences ES positively, and in the direct relationships, CD (contents and materials, teaching strategies, and feedback and assessment) did not influence ES positively. (ii) in the direct relationships, CD (curriculum objectives, contents and materials, teaching strategies, and feedback and assessment) influences ICT knowledge positively. ICT knowledge influences ES positively and mediates between CD and ES. CD (curriculum objectives, contents and materials, teaching strategies, and feedback and assessment) has an indirect relationship with ES through knowledge of ICT.

This study provides the following conclusions based on its findings. First, effective CD plays a significant role in developing ES among students. Second, CD is an antecedent of ICT knowledge and is vital in organizing the knowledge, skills, and capabilities that students should have in the ICT era. Third, ICT knowledge is equally essential for developing ES, such as critical-thinking, problem-solving, risk-taking, and innovation skills. ICT knowledge mediates the

relationship between CD and ES; efficient CD is a positive predictor of including ICT knowledge and competencies, leading to ES. Moreover, this study enhances our understanding of CD implications for the development of ES. There is a need for re-examined curriculum design and delivery and exploring the dependencies between ES and ICT capabilities. In addressing the critical challenges to developing entrepreneurial knowledge, skills, and abilities, the results reflect how CD impacts ICT knowledge and the development of ES and abilities.

## Implications and limitations

### Implications

The results of this study imply that curriculum designers should be striving to enhance delivery more effectively and efficiently toward developing ES. First, curriculum developers should focus on changing the contents, materials, and teaching strategies when redesigning CD; this may also have a significant impact on aligning CD with the development of ES. Second, university leadership, management, policymakers, and curriculum designers should redesign integrated CD (contents, materials, and teaching strategies) across the disciplines related to ES.

### Limitations and future research

The present research has many strengths, but it also has some limitations that may influence the generalization of its findings. First, the participants were from one developing country (Pakistan), which could have led to cultural and economic bias, thereby limiting the generalizability of the research outcomes; similar empirical evidence from other cultural and economic contexts will be of great value in validating the findings. Second, the data were collected from undergraduate, graduate, and postgraduate students; future studies should consider students in different fields and at different levels, undergraduate, graduate, and postgraduate.

The findings and limitations suggest directions for future research. First, future studies should explore other variables such as (i) campus learning environment, (ii) campus affordances, and (iii) entrepreneurial innovation knowledge in the field of entrepreneurship education to evaluate the level of entrepreneurial skills and competencies among students in higher education. Second, future studies should focus on additional factors such as the university learning environment as a mediator to improve the role of CD in entrepreneurial competencies. Finally, we used a cross-sectional research design, but in future a longitudinal study might provide some new understanding of how HE affects the ES of university students.

## Supporting information

**S1 File.**
(XLSX)

## Author Contributions

**Conceptualization:** Javed Iqbal, Muhammad Azeem Ashraf.

**Data curation:** Xie Yi.

**Formal analysis:** Javed Iqbal.

**Funding acquisition:** Muhammad Azeem Ashraf, Ruihua Chen.

**Methodology:** Javed Iqbal, Xie Yi, Muhammad Azeem Ashraf.

**Project administration:** Muhammad Azeem Ashraf.

**Resources:** Shahnaz Perveen, Zahid Imran.

**Software:** Javed Iqbal.

**Supervision:** Muhammad Azeem Ashraf.

**Validation:** Xie Yi, Ruihua Chen, Jin Ning.

**Visualization:** Xie Yi.

**Writing – original draft:** Javed Iqbal, Muhammad Azeem Ashraf.

**Writing – review & editing:** Javed Iqbal, Xie Yi, Muhammad Azeem Ashraf, Ruihua Chen, Jin Ning, Shahnaz Perveen, Zahid Imran.

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
