## [Decision Letter · Decision Letter 0]

2 Dec 2021

PONE-D-21-16858How curriculum delivery translates into entrepreneurial skills: The mediating role of knowledge of information and communication technologyPLOS ONE

Dear Dr. Ashraf,

Thank you for submitting your manuscript to PLOS ONE. After careful consideration, we feel that it has merit but does not fully meet PLOS ONE’s publication criteria as it currently stands. Therefore, we invite you to submit a revised version of the manuscript that addresses the points raised during the review process.

We look forward to receiving your revised manuscript.

Kind regards,

Alessandro Margherita

Academic Editor

PLOS ONE

Journal Requirements:

2. Please improve statistical reporting and refer to p-values as "p<.001" instead of "p=.000". Our statistical reporting guidelines are available at https://journals.plos.org/plosone/s/submission-guidelines#loc-statistical-reporting

3. Please ensure that you refer to Figure 2 in your text as, if accepted, production will need this reference to link the reader to the figure.

4. Please upload a new copy of Figures 1, 2, and 3 as the detail is not clear. Please follow the link for more information: " ext-link-type="uri" xlink:type="simple">https://blogs.plos.org/plos/2019/06/looking-good-tips-for-creating-your-plos-figures-graphics/"
" ext-link-type="uri" xlink:type="simple">https://blogs.plos.org/plos/2019/06/looking-good-tips-for-creating-your-plos-figures-graphics/"

5. Please include a copy of Table 6 which you refer to in your text on page 21.

Reviewers' comments:

Reviewer's Responses to Questions

**Comments to the Author**

1. Is the manuscript technically sound, and do the data support the conclusions?

Reviewer #1: Yes

Reviewer #2: Partly

2. Has the statistical analysis been performed appropriately and rigorously? 

Reviewer #1: I Don't Know

Reviewer #2: Yes

3. Have the authors made all data underlying the findings in their manuscript fully available?

Reviewer #1: No

Reviewer #2: No

4. Is the manuscript presented in an intelligible fashion and written in standard English?

Reviewer #1: Yes

Reviewer #2: No

5. Review Comments to the Author

Reviewer #1: The paper aims to show how curriculum delivery translates into entrepreneurial skills and uses dimensions of curriculum delivery and a questionnaire survey to assess knowledge of information and communication technology and entrepreneurial skills with a study group of 482 students at universities in Lahore, Pakistan. The paper is well-written and easy to follow. Authors claim novelty in analysing the effects of dimensions of Curriculum Development (objectives, contents, teaching strategies and feedback/assessment) with the aim to develop recommendations for administrators and policy makers.

1. Will the team make the data available in a suitable form on open source? If not, please add comments to explain why

2. Methodology: The authors present a good overview of the developed hypotheses and the data collecting, incl. the development of the questionnaire with reference to the original works. The authors use well-documented dimensions of ES, CD and ICT knowledge and justify their selection.

The sample demographics are discussed, however models do not consider personal or environmental factors. Furthermore, sample size is not discussed. Authors should add comments to clarify

3. Please review the document's literature reviews to make sure that it is clear throughout when authors are speaking about ES and ICT knowledge of students vs. ES and ICT knowledge of educators. This can be done e.g. in line 154: "CD can increase ES in students through ICT knowledge of educators"

4. Chapter limitations and future research, should be extended slightly. E.g. line 552: "explore other dimensions of variables", it is not clear what is meant here. Do the authors mean by considering the determinants of skills such as social and or demographic (The authors comment on the demographics of the survey group, but it is not clear if the sample is representative.) or do they refer to including other aspects of ICT literacy? Compared to ICT knowledge used here, the digital skills concept is broader (e.g. Van Laar et al., 2017). They may also want to comment on the effects the authors may expect on the study data and it’s conclusions.

5. Line 534 ff (implications and limitations): Authors should comment on using direct vs. indirect relationships and what the conclusions on the reliability of results are and what this means for the implications.

Very minor:

Line 55: Therefore, we predict that effective CD is associated positively with ES, (predict → hypothesize?)

Line 163: Please check: "This study also complements previous work by clarifying CD in shaping ICT knowledge that leads to CD."

Reviewer #2: 1) Support better the research gap, already in the introduction, with updated and authoritative literature;

2) Provide more details on the characteristics of the research sample, including limitations for generalization;

3) Extend the discussion with theoretical advancements (i.e. how the paper advances the extant knowledge in the field);

4) Replace the figures with high-quality ones;

5) Have the paper proofread by a professional mothertongue.

6. PLOS authors have the option to publish the peer review history of their article (what does this mean?). If published, this will include your full peer review and any attached files.

Reviewer #1: No

Reviewer #2: No

---

## [Author Response · Author response to Decision Letter 0]

3 Mar 2022

PONE-D-21-16858

How curriculum delivery translates into entrepreneurial skills: The mediating role of knowledge of information and communication technology

PLOS ONE

Reviewers' comments:

Reviewer's Responses to Questions

Comments to the Author

1. Is the manuscript technically sound, and do the data support the conclusions?

Reviewer #1: Yes

Reviewer #2: Partly

2. Has the statistical analysis been performed appropriately and rigorously?

Reviewer #1: I Don't Know

Reviewer #2: Yes

3. Have the authors made all data underlying the findings in their manuscript fully available?

The requires authors to make all data underlying the findings described in their manuscript fully available without restriction, with rare exception (please refer to the Data Availability Statement in the manuscript PDF file). The data should be provided as part of the manuscript or its supporting information, or deposited to a public repository. For example, in addition to summary statistics, the data points behind means, medians and variance measures should be available. If there are restrictions on publicly sharing data—e.g. participant privacy or use of data from a third party—those must be specified.

Reviewer #1: No

Reviewer #2: No

4. Is the manuscript presented in an intelligible fashion and written in standard English?

Reviewer #1: Yes

Reviewer #2: No

Answer: We thank the reviewers for their comments. We hired “Charlesworth Author Services” for language check, and we believe the language of the article is checked by the professional editor as well as by native speaker. A certificate of editing is attached. 

5. Review Comments to the Author

Reviewer #1: The paper aims to show how curriculum delivery translates into entrepreneurial skills and uses dimensions of curriculum delivery and a questionnaire survey to assess knowledge of information and communication technology and entrepreneurial skills with a study group of 482 students at universities in Lahore, Pakistan. The paper is well-written and easy to follow. Authors claim novelty in analysing the effects of dimensions of Curriculum Development (objectives, contents, teaching strategies and feedback/assessment) with the aim to develop recommendations for administrators and policy makers.

Comment 1. Will the team make the data available in a suitable form on open source? If not, please add comments to explain why

Answer: we have provided data as supporting documents. 

Comment 2. Methodology: The authors present a good overview of the developed hypotheses and the data collecting, incl. the development of the questionnaire with reference to the original works. The authors use well-documented dimensions of ES, CD and ICT knowledge and justify their selection.

The sample demographics are discussed, however models do not consider personal or environmental factors. Furthermore, sample size is not discussed. Authors should add comments to clarify

Answer: we have revised our article, and we have added sample size, and other clarification on why different factors were not discussed. 

Comment 3. Please review the document's literature reviews to make sure that it is clear throughout when authors are speaking about ES and ICT knowledge of students vs. ES and ICT knowledge of educators. This can be done e.g. in line 154: "CD can increase ES in students through ICT knowledge of educators"

Answer: We thank the reviewer for pointing out this. We have added the required information. 

Comment 4. Chapter limitations and future research, should be extended slightly. E.g. line 552: "explore other dimensions of variables", it is not clear what is meant here. Do the authors mean by considering the determinants of skills such as social and or demographic (The authors comment on the demographics of the survey group, but it is not clear if the sample is representative.) or do they refer to including other aspects of ICT literacy? Compared to ICT knowledge used here, the digital skills concept is broader (e.g. Van Laar et al., 2017). They may also want to comment on the effects the authors may expect on the study data and it’s conclusions.

Answer: We have revised and extended it. 

Comment 5. Line 534 ff (implications and limitations): Authors should comment on using direct vs. indirect relationships and what the conclusions on the reliability of results are and what this means for the implications.

Answer: We thank the reviewer for this suggestion. We have revised this section. 

Very minor:

Line 55: Therefore, we predict that effective CD is associated positively with ES, (predict → hypothesize?)

Line 163: Please check: "This study also complements previous work by clarifying CD in shaping ICT knowledge that leads to CD."

Answer: We appreciate reviewers’ suggestion, and we have revised it in accordance with the comments and suggestions. 

Reviewer #2: 

1) Support better the research gap, already in the introduction, with updated and authoritative literature;

Answer: we thank the reviewer for the suggestion. We have already added it in the introduction section. 

2) Provide more details on the characteristics of the research sample, including limitations for generalization;

Answer: We have revised it, and added research sample. 

3) Extend the discussion with theoretical advancements (i.e. how the paper advances the extant knowledge in the field);

Answer: we have spent a lot of time in writing the revising the discussion, and we appreciate these comments by the reviewer, as it helped us in improving the discussion session 

4) Replace the figures with high-quality ones;

Answer: we appreciate this concern, we have uploaded figure with high quality pixels. 

5) Have the paper proofread by a professional mothertongue.

Answer: For this article, we hired “Charlesworth Author Services” for language check, and we believe the language of the article is checked by the professional editor as well as by native speaker. 

6. PLOS authors have the option to publish the peer review history of their article (what does this mean?). If published, this will include your full peer review and any attached files.

Do you want your identity to be public for this peer review? For information about this choice, including consent withdrawal, please see our Privacy Policy.

Reviewer #1: No

Reviewer #2: No

--

Azeem

---

## [Editor Report · Decision Letter 1]

10 Mar 2022

How curriculum delivery translates into entrepreneurial skills: The mediating role of knowledge of information and communication technology

PONE-D-21-16858R1

Dear Dr. Ashraf,

We’re pleased to inform you that your manuscript has been judged scientifically suitable for publication and will be formally accepted for publication once it meets all outstanding technical requirements.

Kind regards,

Alessandro Margherita

Academic Editor

PLOS ONE
---

## [Editor Report · Acceptance letter]

2 May 2022

PONE-D-21-16858R1 

How curriculum delivery translates into entrepreneurial skills:
The mediating role of knowledge of information and communication technology 

Dear Dr. Ashraf:

I'm pleased to inform you that your manuscript has been deemed suitable for publication in PLOS ONE. Congratulations! Your manuscript is now with our production department. 

Kind regards, 

on behalf of

Dr. Alessandro Margherita 

Academic Editor

PLOS ONE